Morphology and ontogeny of carpus and tarsus in stereospondylomorph temnospondyls

Witzmann Florian 1
Fröbisch Nadia 1 2 nadia.froebisch@mfn.berlin
1 Museum für Naturkunde Berlin, Leibniz Institute for Evolution and Biodiversity Science , Berlin , Germany
2 Department of Biology, Humboldt Universität Berlin , Berlin , Germany
Hutchinson John
Electronic publication date: 2023 Oct 26
Publication date: 2023
Volume: 11
Electronic Location ID: e16182
Received 2023 Mar 13; Accepted 2023 Sep 5
Copyright: © 2023 Witzmann and Fröbisch
Copyright year: 2023
Copyright holder: Witzmann and Fröbisch
License: This is an open access article distributed under the terms of the Creative Commons Attribution License, which permits unrestricted use, distribution, reproduction and adaptation in any medium and for any purpose provided that it is properly attributed. For attribution, the original author(s), title, publication source (PeerJ) and either DOI or URL of the article must be cited.
License URL: https://creativecommons.org/licenses/by/4.0/

Keywords: Limb development, Ossification, Carpus, Tarsus, Tetrapod, Postaxial, Preaxial, Stereospondyl

Funding: German Research Foundation (DFG) Emmy Noether Grant FR2647/5-1 Museum für Naturkunde Berlin This work was funded by the German Research Foundation (DFG) Emmy Noether grant FR2647/5-1 and the Museum für Naturkunde Berlin. The funders had no role in study design, data collection and analysis, decision to publish, or preparation of the manuscript.

==============================
Skeletal development is well known in temnospondyls, the most diverse group of Paleozoic and Mesozoic amphibians. However, the elements of carpus and tarsus (i.e., the mesopodium) were always the last bones to ossify relative to the other limb bones and with regard to the rest of the skeleton, and are preserved only in rare cases. Thus, in contrast to the other parts of the limb skeleton, little is known about the ontogeny and sequence of ossification of the temnospondyl carpus and tarsus. We intended to close this gap by studying the ontogenies of a number of Permo/Carboniferous stereospondylomorphs, the only temnospondyls with preserved growth series in which the successive ossification of carpals and tarsals can be traced. Studying the degree of mesopodial ossification within the same species show that it is not necessarily correlated with body size. This indicates that individual age rather than size determined the degree of mesopodial ossification in stereospondylomorphs and that the largest individuals are not necessarily the oldest ones. In the stereospondylomorph tarsus, the distal tarsals show preaxial development in accordance with most early tetrapods and salamanders. However, the more proximal mesopodials exhibit postaxial dominance, i.e., the preaxial column (tibiale, centrale 1) consistently started to ossify after the central column (centralia 2–4, intermedium) and the postaxial column (fibulare). Likewise, we observed preaxial development of the distal carpals in the stereospondylomorph carpus, as in most early tetrapods for which a statement can be made. However, in contrast to the tarsus, the more proximal carpals were formed by preaxial development, i.e., the preaxial column (radiale, centrale 1) ossified after the central column (centralia 2–4, intermedium) and before the postaxial column (ulnare). This pattern is unique among known early tetrapods and occurs only in certain extant salamanders. Furthermore, ossification proceeded from distal to proximal in the central column of the stereospondylomorph carpus, whereas the ossification advanced from proximal to distal in the central column of the tarsus. Despite these differences, a general ossification pattern that started from proximolateral (intermedium or centrale 4) to mediodistal (distal tarsal and carpal 1) roughly in a diagonal line is common to all stereospondylomorph mesopodials investigated. This pattern might basically reflect the alignment of stress within the mesopodium during locomotion. Our observations might point to a greater variability in the development of the mesopodium in stereospondylomorphs and probably other early tetrapods than in most extant tetrapods, possibly mirroring a similar variation as seen in the early phases of skeletogenesis in salamander carpus and tarsus.

Introduction

The initial evolution of the tetrapod limb and its subsequent role in the diversification of four-limbed vertebrates has been a major research focus in the past decades and new discoveries and thorough redescriptions of the appendicular skeleton in Devonian stem-tetrapods have greatly increased our knowledge about the evolutionary transition from fins to limbs (Coates, 1991, 1996; Coates & Clack, 1990; Boisvert, Mark-Kurik & Ahlberg, 2008; Shubin, Daeschler & Jenkins, 2006; Hall, 2007; Cloutier et al., 2020). At the same time limb development throughout the ontogeny of tetrapods has been studied extensively in order to understand the genetic underpinnings that were involved in the fin-to-limb transition (e.g., Schneider & Shubin, 2013; Wagner & Chiu, 2001, 2003; Woltering & Duboule, 2010). The modular build of the tetrapod limb make it ideal for scrutinizing limb development in different modern tetrapod taxa and led to a large body of data involved in limb patterning and outgrowth (Bénazet & Zeller, 2009; Zeller, Lopez-Rios & Zuniga, 2009; Petit, Sears & Ahituv, 2017; Tickle & Towers, 2017). Paleontological and developmental data have informed and advanced each other in this process and in the past decades the tetrapod limb has become one of the best examples of how insights from different fields, including morphology, paleontology and developmental biology, have come together to gain a broad understanding of the connections between the development and evolution of an organ (Shubin, Tabin & Carroll, 1997; Shubin, 2002; Shubin, Tabin & Carroll, 2009; Schneider & Shubin, 2013).

While all tetrapods share the common ground plan for their limbs and the limb undoubtedly is homologous in all tetrapods, data from fossil taxa as well as from extant non-model organisms provide some indication that certain patterns long believed to be rather constrained and conservative may indeed be evolutionarily more complex and flexible than previously appreciated. This refers in particular to late events in limb development when mesenchymal condensations and subsequent chondrification and ossification form the characteristic elements of the limb skeleton. The vast majority of extant tetrapods, namely frogs and all amniotes, build their limb skeleton from proximal to distal and show a so-called postaxial polarity, especially in the formation of their digits (Fig. 1). Salamanders are a notable exception from this conservative pattern and show a number of differences from it, including 1. preaxial polarity in the formation of the zeugopodial elements and digital arch and 2. a very early condensation of a distal carpal/tarsal element, the basale commune, at the base of digits 1 and 2 (Fig. 1).

Figure 1 Generalized schematic representation of the pattern of skeletogenesis in amniotes/frogs and salamanders.

Darker elements depict already formed skeletal parts with skeletogenesis proceeding from left to right in the figure. Note reversed polarity in the establishment of the zeugopodial elements and the digits in salamanders compared to amniotes and frogs.

When considering extant tetrapods only, this preaxial pattern of limb development seen in salamanders, appears exceptional amongst tetrapods and hence phylogenetically was often considered to be derived (Holmgren, 1933; Wagner et al., 1999; Brockes & Gates, 2014; Kumar et al., 2015). This picture, however, altered when ossification sequences in the limbs of exceptionally well-preserved ontogenetic series of fossil anamniotes from the Paleozoic revealed that preaxial polarity in limb development was not only ancient, but also phylogenetically widespread at least among the large and diverse clade of temnospondyl anamniotes, suggesting that it may indeed be plesiomorphic for tetrapods (Fröbisch, Carroll & Schoch, 2007; Fröbisch, 2008; Fröbisch & Shubin, 2011; Fröbisch et al., 2015). The data from the fossil record has made the picture more complex and it remains thus far elusive when and how often these different patterns of limb development have evolved in the long evolutionary history of tetrapods and what their genetic and/or ecological basis may be.

Moreover, studies on ossification sequences of the limb skeleton have previously focused on elements of the stylopod, zeugopod and the digital arch as these are the parts of the limbs that ossify rather quickly in tetrapod ontogeny and naturally are the elements that are well-preserved in fossil taxa. Contrary, in temnospondyls as well as in many modern amphibian taxa, the elements of the mesopodium (i.e., carpals and tarsals) were always the last bones to ossify relative to the other limb bones and with regard to the rest of the skeleton. Only the pubis started to ossify later in the ontogeny of temnospondyls (Schoch & Witzmann, 2009). Carroll (1997) attributed the delayed ossification of mesopodial elements to different modes of ossification. In contrast to the rest of the limb bones, in which rapid perichondral ossification takes place before the formation of endochondral bone, ossification of carpals and tarsals starts with endochondral bone formation before perichondral ossification occurs. In fact, carpals and tarsals ossified often incompletely or remained completely cartilaginous in a large number of temnospondyls and other early tetrapods. Therefore, and because carpals and tarsals are often small, poorly ossified elements that easily disarticulate, the mesopodium is rarely preserved in temnospondyls, and when, it is often disarticulated and incomplete and the individual elements are difficult to identify. For these reasons, the mesopodium is clearly underrepresented in morphological studies of temnospondyls, where—if preserved—it is mostly casually mentioned. Nevertheless, some exceptions exist which either focused on the temnospondyl mesopodium or provided detailed descriptions within the frame of broader studies, which are summarized in Table 1. While adding a lot of useful morphological information, the listed studies did not address the ontogeny and ossification sequence of the temnospondyl carpus and tarsus. Only Schaeffer (1941, p. 400) shortly mentioned the incompletely ossified tarsus of the early Permian stereospondylomorph temnospondyl Archegosaurus decheni illustrated by von Meyer (1858, pl. XIX, fig. 8) and suggested that ossification started on its proximomedial side. Witzmann (2006), while studying the ontogeny of Archegosaurus, reinvestigated the carpus described by von Meyer (1858) and Jaekel (1909) and illustrated a newly found, incompletely ossified tarsus, and came to the same conclusion as Schaeffer (1941).

Table 1 Previous studies addressing the morphology of temnospondyl mesopodia.

Author/year	Taxon	Comments	
von Meyer (1858)	cf. Archegosaurus decheni	Description of incompletely ossified carpus and tarsalia; pl. VII, fig. 12, pl. XV, fig. 13, pl. XIX, fig. 8	
Quenstedt (1861)	cf. Archegosaurus decheni	Description of almost complete tarsus; fig. 6	
Baur (1886)	cf. Archegosaurus decheni	Interpretation of tarsal bones based on the illustration of Quenstedt (1861)	
Cope (1888)	Eryops megacephalus	Description of complete hand, including complete carpus; plate, fig. 1	
Zwick (1898)	cf. Archegosaurus decheni	Reinvestigation of Quenstedt’s (1861) original specimen and disagreement with some of Baur’s (1886) conclusions; figs. 4, 5	
Emery (1897)	cf. Archegosaurus decheni,
Eryops megacephalus	Discussed and reinterpreted the morphology of the Eryops carpus (fig. 3) based on Cope’s (1888) description and the complete Archegosaurus tarsus based on the works of Quenstedt (1861), Baur (1886) and Zwick (1898) (figs. 4, 5); compared both to extant amphibians and amniotes	
Fraas (1889)	Mastodonsaurus giganteus	Description and illustration of partially preserved carpus; pl. IV, fig. 5	
Jaekel (1909)	cf. Archegosaurus decheni	Comparative morphological survey of the extremities in a range of Paleozoic and Triassic tetrapods including the mesopodium; provided a further reinterpretation of Quenstedt’s (1861) cf. Archegosaurus tarsus (fig. 4), based on a new sketch that Prof. Ernst Koken from Tübingen in Germany had drawn and sent to Jaekel; gave a new illustration (fig. 5) of the incompletely ossified carpus first described by von Meyer (1858)	
Williston (1909)	Acheloma cumminsi	Description of complete tarsus; fig. 5 (Williston used the old name Trematops milleri, which is not valid; see Dilkes & Reisz, 1987)	
von Huene (1922)	Eryops megacephalus	Redescription of the Eryops carpus first published by Cope (1888); fig. 44	
Gregory, Miner & Noble (1923)	Eryops megacephalus	Redescription of the Eryops carpus first published by Cope (1888); fig. 1	
Miner (1925)	Eryops megacephalus	Redescription and illustration of the Eryops carpus first published by Cope (1888); fig. 21	
Schaeffer (1941)	cf. Archegosaurus decheni,
Acheloma cumminsi	Description of tarsus of Acheloma (fig. 1) and reinterpretation of the tarsus of cf. Archegosaurus (fig. 2). Comparisons with further early tetrapods and extant amphibians; concentrated on functional aspects of the amphibian and reptilian tarsus	
Dilkes (2015)	Eryops megacephalus, Acheloma cumminsi, Dissorophus multicinctus, Cacops aspidephorus	Detailed redescriptions of Cope’s (1888) Eryops hand (fig.1) and the hands of Dissorophus (fig. 3) and Acheloma (fig. 4); provided descriptions of the tarsus of Eryops (fig. 6) and Acheloma (fig. 5), as well as of carpus and tarsus of Dissorophus (fig. 7) and isolated carpal and tarsal elements of Cacops (fig. 8); identified several phylogenetically informative mesopodial characters	
Witzmann (2006)	cf. Archegosaurus decheni	Reinvestigated carpus described by von Meyer (1858) and Jaekel (1909) and illustrated a newly found, incompletely ossified tarsus (fig. 4).	

Otherwise, all previous accounts that included the ontogeny of temnospondyl hands and feet focused on phalanges and metapodia (Schoch, 1992; Fröbisch, Carroll & Schoch, 2007; Fröbisch, 2008), simply due to of the rarity of ossified carpi and tarsi. Thus, little is known about the ontogeny of the temnospondyl carpus and tarsus and their ossification sequences in temnospondyls, in contrast to the other parts of the limb skeleton.

With this study we intend to close this gap by studying the ontogenies of a number of basal stereospondylomorph taxa, including the stem-stereospondyls Sclerocephalus, Glanochthon, Cheliderpeton and Archegosaurus, and early stereospondyls like Uranocentrodon, which are the only temnospondyls from which growth series and successive ossification of carpals and tarsals are preserved. More derived stereospondyls have completely cartilaginous mesopodia with rare exceptions of single ossified elements in large individuals of Gerrothorax and Mastodonsaurus (Fraas, 1889; Nilsson, 1946; Schoch, 1999).

Recently, Jia et al. (2022) provided a comprehensive study of the ossification patterns in the mesopodium of a variety of salamander clades and compared their results with published descriptions of incompletely ossified carpals and tarsals of a few selected early tetrapods. The authors found preaxial dominance in the ossification of distal carpals and tarsals in most salamander taxa and suggested this pattern to be plesiomorphic for tetrapods, in line with previous studies suggesting a plesiomorphic state of preaxial polarity in limb development based on ossification of the long bones and digits (Fröbisch, Carroll & Schoch, 2007; Fröbisch, 2008; Fröbisch, Bickelmann & Witzmann, 2014; Fröbisch et al., 2015). The aim of the present study is (1) to provide a morphological reconstruction of carpus and tarsus in basal stereospondylomorphs, (2) to resolve their ontogeny and ossification sequences as far as possible, and (3) to compare the ontogenetic pattern with that of extant lissamphibians and amniotes. Therein, a particular question of interest for this study is, if the ossification sequences of carpus and tarsus reflect pre- and/or postaxial development of the limb skeleton as described above.

Material and Methods

For a list of stereospondylomorph taxa studied either first hand or taken from the literature please see Table 2.

Table 2 Temnospondyl carpi and tarsi examined for the present study.

Taxon	Specimens studied first-hand	Data from the literature	
A. Carpus	
Sclerocephalus haeuseri	SMNS 90055, original to Schoch & Witzmann (2009, fig. 8d); SMNK-Pal 6402		
cf. Archegosaurus decheni	MB.Am.255a, b, original to von Meyer (1858, pl. XV, fig. 12), Jaekel (1909, fig. 5) and Witzmann (2006, fig. 4a)		
Platyoposaurus stuckenbergi		Gubin (1991, fig. 34b)	
Uranocentrodon senekalensis		Van Hoepen (1915, pl. XXI)	
B. Tarsus	
Sclerocephalus nobilis	NHMM-PW 2005/2Ls, original to Schoch & Witzmann (2009, fig. 8e)		
Glanochthon lellbachae	SMNS 91281 (cast: SMNS 91279)		
cf. Archegosaurus decheni	GPIT/AM/781a, b, original to Quenstedt (1861, fig. 6), Zwick (1898, fig. 1) and Jaekel (1909, fig. 4); IGS U II 1/1a, b; IGS U II 3/1, original to Witzmann (2006, fig. 4b)	von Meyer (1858, pl. XV, fig. 13, p. 183, pl. XIX, fig. 8)	
Cheliderpeton vranyi		Fritsch (1889, p. 54); Werneburg & Steyer (2002, fig. 3a)	
Uranocentrodon senekalensis		Van Hoepen (1915, pl. XXII); Broom (1921, fig. 2)	
Note:

For abbreviations, see text.

Further stereospondylomorphs with ossified carpals and/or tarsals which are, however, fragmentarily preserved or inadequately known are Australerpeton cosgriffi (Dias & Schultz, 2003), Lydekkerina huxleyi (Pawley & Warren, 2005; Hewison, 2008), Gerrothorax pulcherrimus (Nilsson, 1946) and Mastodonsaurus giganteus (Fraas, 1889; Schoch, 1999). They are not considered here.

At this point, it is appropriate to give a short note concerning the potential sources of error that may arise during the investigation of ontogenies in the fossil record, but which normally do not occur in studies of developmental patterns in extant vertebrates. First, apart from being rarely preserved per se, ossification sequences are naturally incomplete in fossils (Fröbisch et al., 2010). The paleontologist has to be careful in the interpretation of the presence or absence of certain bones in the differently sized skeletons: is the particular element missing because it was still cartilaginous when the animal died, or is it just an artefact of preservation? Careful consideration of the taphonomic conditions is necessary in each case. Similar care is required with respect to the second source of error, the possible postmortem alterations of the morphology of skeletal elements and their spatial relationships to each other (especially when ossification proceeded very slowly as in mesopodial bones of temnospondyls). Third, to interpret a fossil size series as growth series, it has to be ensured that the differently sized specimens belong to the same species and are derived from the same locality and ideally from the same horizon (Schultze, 1984). Otherwise, microevolutionary changes and geographical variation might influence the reconstruction of a fossil ontogeny (Fröbisch et al., 2010). In the following description and interpretation of temnospondyl mesopodia, we will refer to these limitations where necessary.

The directional terms used in the following description are based on Dilkes (2015). “Dorsal” indicates the front or upper side of carpus and tarsus, whereas “ventral” describes the palmar (carpus) and plantar (tarsus) side, respectively. “Proximal” refers to the portion of carpus and tarsus closer to radius/ulna and tibia/fibula, respectively, and “distal” closer to metacarpals and–tarsals, respectively. “Medial” refers to the direction to the first digit (preaxial), and “lateral” to the direction to the last digit (postaxial) (Fig. 2). Furthermore, the “length” of an element refers to its proximodistal extension, and the “width” is perpendicular to it.

Figure 2 Eryops megacephalus, the classical study object of the temnospondyl mesopodium.

(A) Carpus; (B) tarsus. Redrawn from Dilkes (2015, figs. 2a, 6d). The bones of the preaxial column are held in blue, the central column in yellow, the postaxial column in purple and the distal carpals and tarsals, respectively, in red. Abbreviations: c, centrale; dc, distal carpal; dt, distal tarsal; int, intermedium; mc, metacarpal; mt, metatarsal; r, radius; rad, radiale; u, ulna; I–V, metacarpals and metatarsals, respectively; (C) left forelimb of a cleared and stained larval Cryprobranchus alleganiensis, blue are cartilaginous structures, red are ossified elements, orientation as in (A) and (B).

To compare the mesopodial ossification patterns in stereospondylomorphs among each other, but also with other temnospondyls, salamanders and amniotes, we follow Jia et al. (2022) who used the scheme of Goette (1879) in dividing the salamander mesopodium into three different columns. These columns contain, in the case of a “generalized” salamander mesopodium like that of Dicamptodon (Shubin & Wake, 2003), the following elements: (1) radiale/tibiale and element y in the preaxial column, (2) intermedium and centrale in the central column, and (3) ulnare/fibulare in the postaxial column. Compared to temnospondyls, however, salamanders have reduced the number of their carpal and tarsal elements to varying degrees by loss or fusion (Shubin & Alberch, 1986; Shubin & Wake, 2003). Therefore, applying Goette’s (1879) scheme to temnospondyls with their larger number of mesopodial ossifications, this means that (1) the preaxial column contains the radiale/tibiale and centrale 1 (which corresponds to element y in salamanders, see Schaeffer (1941) and Dilkes (2015)), (2) the central column corresponds to the intermedium, centrale 4 (the centrale of salamanders, see Schaeffer (1941) and Dilkes (2015)), centrale 3, and centrale 2, and (3) the postaxial column consisting of ulnare/fibulare (as in salamanders) (Fig. 2). The row of distal carpals and distal tarsals, respectively, or digital arch mesopodials sensu Jia et al. (2022), are treated separately from these columns.

Repository and institutional abbreviations–GPIT, Paläontologische Sammlung der Universität Tübingen; MB, Museum für Naturkunde Berlin; NHMM-PW, Naturhistorisches Museum Mainz; SMNK, Staatliches Museum für Naturkunde Karlsruhe; SMNS, Staatliches Museum für Naturkunde Stuttgart.

Results

General morphology of the temnospondyl mesopodium

The classic study objects of the temnospondyl carpus and tarsus have been Eryops megacephalus and Acheloma cumminsi (formerly Trematops milleri) (Cope, 1888; von Huene, 1922; Gregory, Miner & Noble, 1923; Miner, 1925; Schaeffer, 1941; Dilkes, 2015). Based on their morphologies, the plesiomorphic state for the temnospondyl mesopodia has been suggested as consisting of radiale, intermedium and ulnare proximally, four centralia in the middle part of the carpus, and finally four distal carpalia for the carpus (Fig. 2A), and of tibiale, intermedium, fibulare, four centralia and five distal tarsalia for the tarsus (Fig. 2B). This has been confirmed in the meantime by the mesopodia of less derived temnospondyls like the edopoid Cochleosaurus bohemicus (Sequeira, 2009) and the “dendrerpetids” Balanerpeton woodi (Milner & Sequeira, 1994) and probably Dendrerpeton acadianum (Dilkes, 2015) who have the same number and identity of mesopodial elements. This pattern was obviously already established in post-Devonan stem-tetrapods, as indicated by the well-ossified tarsus in the colosteid Greererpeton burkemorani (Godfrey, 1989; Coates & Ruta, 2007). In contrast, Devonian stem-tetrapods have a variable number of tarsal and carpal elements whose homology is often uncertain (Coates & Ruta, 2007; Carroll & Holmes, 2007). In this paper, all carpi and tarsi are drawn with the preaxial column to the left to facilitate comparisons between specimens.

Description of the carpus in stereospondylomorphs

Among stereospondylomorphs, only a few specimens with ossified carpal elements are known, and in most of them the carpus is either incompletely ossified (e.g., Sclerocephalus haeuseri, cf. Archegosaurus decheni, Glanochthon lellbachae, Uranocentrodon senekalensis) or only fragmentarily preserved (e.g., Platyoposaurus stuckenbergi, Lydekkerina huxleyi, Mastodonsaurus giganteus). The carpus is more rarely preserved in stereospondylomorphs than the tarsus, probably because it started to ossify later in ontogeny (Boy, 1988; Meckert, 1993; Witzmann, 2006; Schoch & Witzmann, 2009).

Cf. Archegosaurus decheni

A single, large carpus from Lebach (MB.Am.255a, b) has been described by several authors (von Meyer, 1858; Jaekel, 1909; Witzmann, 2006; Witzmann & Schoch, 2006) and can either be ascribed to Archegosaurus decheni or Glanochthon latirostris, the two stereospondylomorphs that occur in this locality. Since G. latirostris is a rather rare component of the Lebach fauna, this specimen is tentatively designated as cf. Archegosaurus decheni. It is redescribed in the following. MB.Am.255 is a partially ossified, right carpus with metacarpalia and phalanges and associated with radius and ulna. According to its size, it must have belonged to an exceptionally large (and thus probably late adult) individual. The carpus consists of six ossified elements (intermedium, radiale, centralia 1, 2, and 4, and distal carpal 1) (Figs. 3A and 3B). Reexamination of a seventh element designated as ulnare by Witzmann (2006) and Witzmann & Schoch (2006) revealed that it is an artefact and not a carpal bone. All ossified elements are rather small and are not in contact with each other, which indicates continuation in cartilage to a large degree. The intermedium is the smallest bone and has a completely unfinished surface. In dorsal view, it is wider than long and of roughly hexagonal shape. It is not preserved in ventral view. Centrale 4 is located mediodistal to the intermedium. It is the largest bone, much wider than long, and has a roughly trapezoidal shape in dorsal and an ovate shape in ventral view. Large parts of the dorsal and ventral surface consist of finished, periosteal bone. The ventral surface is concave and bears a large number of pits. Medial to centrale 4 is the radiale, a square to slightly pentagonal small element of about half the size of centrale 4. Its surface consists completely of unfinished bone. Centrale 2 is located mediodistal to centrale 4, is wider than long, ovate in outline and is the second largest bone in the carpus. Similar to centrale 4, its dorsal and ventral surface consist to a large part of periosteal bone. The distal edge of the bone is lined by a shallow rim dorsally. The mediodistally following centrale 1 is only slightly larger than the intermedium, minimally longer than wide, of irregular outline and its surface consists completely of unfinished bone. The distalmost ossified element, distal carpal 1, is similar in outline to central 2, but slightly smaller and bears no periosteal bone.

Figure 3 Incompletely ossified hand of a large specimen of cf. Archegosaurus decheni (MB.Am.255a, b).

Among the ossified carpal elements, only centralia 4 and 2 show smooth periosteal bone. The drawings were made based on silicone casts. (A) Dorsal view (MB.Am.255a); (B) ventral view (MB.Am.255b). The intermedium is not preserved in ventral aspect. Abbreviations: c, centrale; dc, distal carpal; int, intermedium; mc, metacarpal; phal, phalange; r, radius; rad, radiale; u, ulna; I–III, metacarpals.

An interesting observation in this specimen is that some carpals had developed periosteal bone (centralia 2 and 4) whereas in all other elements the entire bone surface is unfinished. Because periosteal bone is formed after endochondral bone formation in mesopodial elements (Rieppel, 1992; Carroll, 1997), it can be assumed that centralia 2 and 4 were the first carpal elements that started to ossify in cf. Archegosaurus decheni.

Sclerocephalus haeuseri

Completely ossified carpus.–To our best knowledge, the only completely ossified and preserved stereospondylomorph carpus is present in the left forelimb of Sclerocephalus haeuseri SMNK-Pal 6402 (skull length c. 150 mm) in which, however, preservation of the particular elements is poor, probably because most of the elements were not yet well ossified at the time of death of this individual. In the carpus of SMNK-Pal 6402, the following ossified carpalia are preserved in ventral aspect (Fig. 4A): intermedium, radiale and ulnare, centralia 1, 2, 4 and probably 3, and distal carpalia 2–4. The only missing element is distal carpal 1, and considering the ossification sequence in the stereospondylomorph carpus (see below) and the state of preservation, we consider that an ossified distal carpal 1 was initially present but was lost from the specimen. Except for the intermedium and ulnare, all carpal bones are separated by gaps, indicating that the periphery of the elements still consisted to a large degree of cartilage. The intermedium is the proximalmost bone and is a small element, wider than long, and is roughly pentagonal in shape. Its distal margin is pointed, forming two articulation facets for centrale 4 and the ulnare, respectively. The radiale is a trapezoidal element with a rounded lateral edge. Its proximal straight, oblique margin contacts the mediodistal facet of the radius. Laterally, it contacts centrale 4, distally centrale 1 and laterodistally centrale 2. The ulnare is of roughly semicircular outline, but the concave, more proximolateral part is poorly preserved. The slightly concave proximomedial margin of the ulnare articulated with the intermedium. The irregularly preserved medial edge articulated with centrale 4, and the mediodistal margin with distal carpal 4. Centrale 4 is the largest element of the carpus, probably followed by centrale 2 whose dimensions are not entirely clear. Centrale 4 is diamond-shaped and wider than long. Its proximomedial margin articulates with a corresponding facet on the radius, and its proximolateral margin articulates with the intermedium. Its medial and lateral edges are contact points for the radiale and ulnare, respectively. The rather straight mediodistal and laterodistal margins articulate with centrale 2 and distal carpal 4, respectively, and the distal tip of centrale 4 probably contacted centrale 3. Centrale 2 is poorly preserved but can be reconstructed as wider than long. Its straight mediodistal margin articulated with centrale 1 and the radiale, and its distal facet with distal carpal 2. Laterodistally, it contacted distal carpal 3, and its small lateral facet articulated probably with centrale 3. Proximolaterally, it articulated with centrale 4. Because of poor preservation, centrale 3 cannot be demonstrated unequivocally; if our interpretation is correct, then it is the smallest preserved carpal element and located between centralia 2 and 4 and distal carpals 3 and 4. Also centrale 1 is poorly preserved and represented only by its proximal margin. Apparently, it was only slightly larger than centrale 3 and articulated proximally with the radiale, laterally with centrale 2 and distally probably with the missing distal carpal 1. Distal carpals 2–4 are of poorly defined, irregular outline, probably because they were still poorly ossified. Distal carpal 2 is located between metacarpal II and a straight, distal facet of centrale 2. As preserved, distal carpal 3 is more slender than distal carpals 2 and 4. It articulates with metacarpal III, the oblique, straight margin of centrale 2 and probably centrale 3. Apart from metacarpal IV, distal carpal 4 articulated with distal carpal 3, probably centrale 3, and centrale 4 and the ulnare.

Figure 4 Carpi of different stereospondylomorph temnospondyls.

(A) Almost completely ossified carpus of Sclerocephalus haeuseri (SMNK-Pal 6402). Distal carpal 1 was probably ossified but was lost from the specimen; (B) incompletely ossified carpus of S. haeuseri (SMNS 90055; redrawn after Schoch & Witzmann, 2009, fig. 8d); (C) distal fragment of a probably completely ossified carpus of Platyoposaurus stuckenbergi (redrawn after Gubin, 1991, fig. 34b), the arrow points to the break line through centrale 4; (D) incompletely ossified carpus of Uranocentrodon senekalensis (redrawn after Van Hoepen, 1915, pl. XXI). A, B, and D are reversed to facilitate comparison with other specimens. The bones of the preaxial column are held in blue, the central column in yellow, the postaxial column in purple and the distal carpals and tarsals, respectively, in red. Abbreviations: c, centrale; dc, distal carpal; int, intermedium; mc, metacarpal; phal, phalange; r, radius; rad, radiale; u, ulna; uln, ulnare; I–IV, metacarpals.

Incompletely ossified carpi.–In Sclerocephalus haeuseri, no ossified carpal elements are known in specimens with a skull length smaller than 140 mm (Meckert, 1993). Schoch & Witzmann (2009, fig. 8d) so far provided the only illustration of an incompletely ossified carpus of S. haeuseri. It belongs to an adult specimen (SMNS 90055) with a skull length of 198 mm and is preserved in dorsal view (Fig. 4B). Five carpals are ossified, the radiale, centralia 4, 2, and 1, and distal carpal 1. A small piece of bone distal to distal carpal 1 may be broken off from this bone or represent a sesamoid bone. The large proportion of unfinished bone and the comparatively large distance between some of the elements indicates that they were continued to a large extent in cartilage. This is supported also by the roundish shape of the elements. The only exception is the largest bone, centrale 4, which has a pentagonal outline. The radiale and centrale 2 are roughly of the same size, followed by distal carpal 1 and centrale 1, which is the smallest element.

Only Meckert (1993, p. 128) described another carpus of an adult specimen of S. haeuseri, but did not illustrate it. According to this author, the carpus consists of a square radiale, a pentagonal centrale 4 and an equally sized, elongate ulnare. Additionally, Meckert mentioned the presence of centralia 1 and 2, but the remaining elements could not be identified due to poor preservation.

Platyoposaurus stuckenbergi

Apart from Sclerocephalus haeuseri-specimen SMNK-Pal 6402 (see above), the only other apparently completely ossified stereospondylomorph carpus that we are aware of is that of Platyoposaurus stuckenbergi as illustrated by Gubin (1991, fig. 34b), albeit its proximal part is broken off (Fig. 4C). In this specimen, all carpal bones except for radiale, intermedium and centrale 1 are preserved. Similar to SMNK-Pal 6402, both centralia 4 and 2 (mediale and centrale 1 sensu Gubin, 1991) are large and wider than long, whereas centrale 3 (centrale 2 sensu Gubin, 1991) is a tiny element. The ulnare is similar to that of SMNK-Pal 6402 in being crescentic in outline, but it is proximodistally elongate. A further similarity might be that distal carpal 3 is more slender than distal carpals 2 and 4. Distal carpal 1 in P. stuckenbergi is proximodistally elongate and slender. The large gap in SMNK-Pal 6402 between metacarpal I and centrale 1 might indicate that distal carpal 1 was of similar outline in this specimen.

Uranocentrodon senekalensis

A photograph of an incompletely ossified carpus of the basal stereospondyl Uranocentrodon senekalensis was published by Van Hoepen (1915, pl. XXI). Only two carpalia, a proximal and a more distal one, are ossified in this specimen (Fig. 4D). The proximal, larger element is roughly ovate and much wider than long, but its exact shape cannot be determined based on the photograph. It contacts the mid part of the distal end of the radius. The second, smaller element articulates with the laterodistal surface of the larger element and is also wider than long. Van Hoepen (1915) interpreted the proximal bone as the radiale and the more distal one as distal carpal 1. However, proportional size and location relative to the radius of Van Hoepen’s (1915), radiale corresponds much more to centrale 4 of Sclerocephalus and cf. Archegosaurus. Considering that, his distal carpal 1 is more probably a centrale 2, because it contacts centrale 4, and there is a large gap between this element and the metacarpals what makes identification as a distal carpal unlikely.

Description of the stereospondylomorph tarsus

Cf. Archegosaurus decheni

Completely ossified tarsus.–The almost completely ossified tarsus from Lebach first described by Quenstedt (1861) belongs either to Archegosaurus decheni or Glanochthon latirostris, but as the isolated hand from Lebach described above it is tentatively ascribed to cf. Archegosaurus decheni because this taxon occurs much more frequently in Lebach. After Quenstedt’s (1861) original work, the specimen had been redescribed and discussed by subsequent authors like Baur (1886), Zwick (1898), Emery (1897), Jaekel (1909) and Schaeffer (1941), but no consensus concerning the assignment of the particular tarsal elements was achieved. In their description of the Archegosaurus postcranium, Witzmann & Schoch (2006) reinterpreted the identity of the tarsal bones in this specimen based on the published illustrations and considered this specimen as being lost since it was not detectable in the Tübingen collection at that time. Fortunately, the specimen has been found again in the meantime (inventory number GPIT/AM/781a, b) and is newly described in the following (Figs. 5A and 5B).

Figure 5 Completely ossified tarsus of cf. Archegosaurus decheni (GPIT/AM/781a, b).

(A) Dorsal view (GPIT/AM/781a); (B) ventral view (GPIT/AM/781b). The drawings were made based on silicone casts. Abbreviations: c, centrale; dc, dermal scales; dt, distal tarsal; f, fibula; fib, fibulare; int, intermedium; mt, metatarsal; t, tibia; tib, tibiale; I–IV, metatarsals.

The specimen is preserved as plate (a: dorsal view) and counter plate (b: ventral view) and consists of an almost completely ossified tarsus, four metatarsalia, each of them associated with a proximal phalange, two isolated phalanges, tibia and fibula, and a poorly preserved femur. The tibia measures 48 mm, the fibula 44 mm and metatarsal IV 25 mm. Previous authors did not agree if the four preserved metatarsals represent metarsalia I–IV or II–V. Here we interpret the four metatarsals as I–IV for the following reasons: (1) Metatarsal I is smaller and more slender than the other metatarsals in ontogenetically advanced specimens from Lebach (e.g., von Meyer, 1858, pl. XV, fig. 14; Witzmann, 2006, fig. 4b), and this holds true for the medialmost preserved metatarsal in GPIT/AM/781. (2) Centrale 2 can be determined with confidence in this specimen (see below), and accordingly, the directly adjacent tarsal mediodistal to it can be designated as distal tarsal 1. The metatarsal interpreted here as metatarsal I is located directly distal to this element.

Eleven ossified tarsal elements are present in GPIT/AM/781. This is almost the full complement of a temnospondyl tarsus, only distal tarsal 5 is missing. It was either not ossified or—what we regard as more likely—is simply not preserved (as the 5th metatarsal and the associated phalanges). All bones have finished periosteal bone surfaces on the dorsal and ventral side. The proximal tarsals—intermedium, fibulare, centrale 4 and tibiale—are strikingly similar in morphology and configuration to the tarsus of Eryops megacephalus (Fig. 2B) or Acheloma cumminsi (Dilkes, 2015). The intermedium is a large element that almost reaches the size of the fibulare. It is longer than wide and crescent shaped. Its concave edge is rounded and covered with periosteal bone; it faces medioproximally towards the tibia. Distally, its convex margin fits into the slightly concave facet of centrale 4. There is an almost straight contact with the fibulare laterally, and lateroproximally, the intermedium articulated with the mediodistal facet of the fibula. The fibulare is slightly larger than the intermedium and longer than wide. It is proximo-distally elongate, roughly pentagonal and tapers distally. Medially, it bears a proximal and a distal facet; the proximal one articulates with the intermedium, and the distal one with the lateral side of centrale 4. Proximally the fibulare articulated with the mediodistal part of the fibula, whereas distally it contacted distal tarsal 4 and probably distal tarsal 5. Slight disarticulation has increased the distance between fibulare and fibula and distal tarsal 4, respectively. Centrale 4 is the largest element of the tarsus. It is much longer on the medial than of the lateral side, with distinctly concave proximal and distal margins. The difference in relative length of the two sides is also present—albeit less pronounced—in centrale 4 of Eryops, Acheloma and Dissorophus (Dilkes, 2015). The proximal and distal sides of centrale 4 are likewise concave as in cf. Archegosaurus, and this holds true also for the proximal side of the element in Eryops; in contrast, its distal edge bears only a slight concavity where it contacts distal tarsal 4 (Dilkes, 2015). Centrale 4 in GPIT/AM/781 is wider than long and has a slightly concave dorsal surface. Interestingly, the ventral side is deeply concave in Eryops, Acheloma and Dissorophus (Dilkes, 2015). The proximal concavity accommodates the convex distal margin of the intermedium and the distal one receives centrale 3 and contacted also distal tarsal 4 laterodistally. Mediodistally, it contacts centrale 2, and the proximal half of the straight medial margin articulates with the tibiale. As in Eryops and Acheloma (Dilkes, 2015), the tibiale is a square element, but it appears much abbreviated and is strikingly small. There is a large gap between this bone and centrale 1 (which is normally contacted by the tibiale), which might be an indication of a distal continuation of the tibiale in cartilage. Using Eryops and Acheloma as guides, this is further supported by the presence of a probable contact surface for the tibiale along the complete lateral side of centrale 4 that is longer than the preserved length of the tibia (D. Dilkes, 2023, personal communication). Centrale 2 is pentagonal in outline, with the lateral margin forming two facets for centrale 3 and distal tarsal 3, respectively. Thereby, the contact with the latter bone is more substantial than in Eryops and Acheloma (Dilkes, 2015). It contacts distal tarsals 1 and 2 distally, and centrale 1 medially. Centralia 1 and 3 are small elements, ovate in outline and longer than wide. As preserved, centrale 1 contacts only centrale 2, but there was probably a cartilaginous contact with the tibiale proximally (see above). Apart from centrale 4, centrale 3 articulates with distal tarsal 4 laterally, distal tarsal 3 distally and centrale 2 medially. Distal tarsals 1, 2 and 4 are roughly quadrangular in shape with rounded margins, whereas distal tarsal 3 is wider than long. Distal tarsalia 1 and 2 are smaller than distal tarsalia 3 and 4.

Incompletely ossified tarsi.–As in Sclerocephalus haeuseri, all larvae, juveniles and even subadults of Archegosaurus decheni show no signs of ossified carpal and tarsal elements until a skull length of at least 220 mm (Witzmann & Schoch, 2006). von Meyer (1858, pl. XV, fig. 13) illustrated an isolated foot in which only one tarsal element is ossified (Fig. 6A). The metatarsals and phalanges of four toes are preserved, and they are displaced towards tibia and fibula. The presumed metatarsal IV measures 18 mm. The tarsal bone in question was apparently still poorly ossified as von Meyer (1858, p. 183) described it as dünnen knöchernen Lappen (“thin osseous lobe”). It is comparatively large and its shape is pentagonal with one concave edge. Comparison with the completely ossified tarsus GPIT/AM/781 (and further specimens, see below) indicates that this element represents the intermedium. von Meyer (1858, pl. XIX, fig. 8) also illustrated another isolated foot (tibia length ca. 38 mm, fibula length ca. 36 mm) in which ossification of the tarsus is more advanced (Fig. 6B). It consists of a bony mass, which extends from the mediodistal edge of the fibula in a mediodistal direction. We interpret this bony mass as consisting of two elements, although a boundary between them is not detectable in von Meyer’s figure, probably for taphonomic reasons. If we assume a boundary in the region where the bony mass is somewhat constricted, then we have a proximal element whose shape corresponds closely with the intermedium in other specimens of cf. A. decheni. This interpretation is supported by the location of this element between tibia and fibula. If this interpretation is correct, then the more distal element is centrale 4. Although its margins are irregular (probably because it was in an incipient stage of ossification), its general shape is rhombic and it is wider than long. More difficult to interpret is a smaller bone fragment mediodistal to centrale 4. According to its size and location, it probably represents centrale 2; however, von Meyer (1858, p. 182) assumed that it could be a (displaced) metatarsal. A hindlimb (tibia length 38 mm, metatarsal IV 20 mm) containing a more advanced tarsus with four tarsal ossifications from the palaeontological collection in Strasbourg was illustrated by Witzmann (2006, fig. 4b), encompassing intermedium, centrale 2 and 4 and distal tarsal 1 (Fig. 6C). The bones are properly aligned in one oblique row from the mediodistal edge of the fibula towards metatarsal I. Notably, the particular bones are proportionally much smaller than the respective ones in von Meyer’s figures. All bones have roughly the same size, with centrale 4 being slightly wider and distal tarsal 1 being slightly smaller. Another tarsus from a specimen of the same size (tibia length 38 mm, metatarsal IV 20 mm) but with a distinctly more advanced state of ossification is also housed in the Strasbourg collection (IGS U II 1/1). The specimen consist of a complete hindlimb, the pelvic region and the anterior part of the tail. The tarsus is almost completely ossified, with only the tibiale, centralia 1 and 3, and distal tarsal 5 missing. The particular tarsal bones are in a poor state of preservation, and it is difficult to reconstruct their original outline (Fig. 6D). The intermedium is very similar in outline to that in the completely ossified tarsus GPIT/AM/781 and this holds true also for centrale 4 which is likewise the largest element of the tarsus here. The fibulare has not attained the elongate shape as in GPIT/AM/781, and centrale 2 is broad ovate in shape. It is not clear if a small centrale 3 was ossified. Distal tarsals 1 to 4 are ossified. The first one is almost circular, and the second is smaller and irregular in outline, what might be the result of poor preservation. The third and fourth distal tarsals are broad ovate in outline.

Figure 6 Incompletely ossified tarsi of cf. Archegosaurus decheni.

(A) Redrawn from von Meyer (1858, pl. XV, fig. 13); (B) redrawn from von Meyer, 1858, pl. XIX, fig. 8); (C) redrawn from Witzmann (2006, fig. 4b); (D) IGS U II 1/1b. C is reversed to facilitate comparison with other specimens. The bones of the central column are held in yellow, the postaxial column in purple and the distal carpals and tarsals, respectively, in red. Abbreviations: c, centrale; dt, distal tarsal; f, fibula; fib, fibulare; int, intermedium; mt, metatarsal; t, tibia; I–V, metatarsals.

Sclerocephalus haeuseri

Completely ossified tarsus.–Boy (1988, fig. 10b) provided the so far only illustration of a completely ossified tarsus of Sclerocephalus haeuseri but did not provide a description of it. The configuration of the elements is strikingly similar to the tarsus of Eryops (Fig. 2B), whereas the shape and relative size of particular bones may be different (Fig. 7A). The fibulare is the largest element of the tarsus and has the outline of an inverted, stout “L”. Its proximal, slightly convex end articulates with the mid-part of the distal end of the fibula. Its medial, straight margin articulates with intermedium and centrale 4, and its distal, convex end is received by a notch of distal tarsal 4. Conversely, the laterodistal side of the fibula is concave and receives the proximomedial margin of distal tarsal 5. Between the proximolateral margin of the fibulare and the fibula and the lateral margin of the fibula and distal tarsal five is a small bone in each case. These were identified as sesamoid bones by Boy (1988). The intermedium has about half the size of the fibulare and is kidney-shaped, with the concave side facing medially towards the laterodistal edge of the tibia. This is similar to the intermedium in Eryops (Dilkes, 2015) and cf. Archegosaurus (see above). The distal, slightly convex edge of the intermedium fits into the concave distal margin of centrale 4, whereas the proximal end of the intermedium articulates with the medial facet of the distal end of the fibula. As already mentioned by Boy (1988), the tibiale is strikingly small, longer than wide and trapezoidal in shape. Its proximal, straight margin articulates with the mediodistally facing facet of the tibia, its lateral straight margin with centrale 4, and its short distal margin with centrale 1. Centrale 4 is the second largest element after the fibula and pentagonal in shape. Its concave proximal articulation with the intermedium and straight lateral contact with the tibiale was already mentioned; its lateral margin bears two facets: the more proximal one contacts the fibulare and the more distal one contacts distal tarsal 4. The distal tip of centrale 4 forms point contact with centrale 3. The distal, slightly convex edge of centrale 4 articulates with the concave proximal margin of centrale 2. This crescent-shaped element is much wider than long and contacts the small, square centrale 1 medially, and its distinctly convex margin forms two facets, one for distal tarsal 1 mediodistally and the other one for distal tarsal 2 mediolaterally. Centrale 2 is laterally pointed and establishes point contact with centrale 3, which is the smallest element of the tarsus (comparable with the two sesamoid bones mentioned above) and ovate in shape with a convex proximal and a concave distal margin. It contacts centrale 2 laterally, centrale 4 proximally, distal tarsal 4 laterally and distal tarsal 3 distally. Distal tarsalia 1–3 and 5 are similar in size as the tibiale, with a rather straight or slightly concave distal and a convex proximal margin. Distal tarsal 4 is much larger (similar to Eryops; Dilkes, 2015) and L-shaped. Apart from the proximal ends of the respective metatarsalia and their adjacent distal tarsals, distal tarsal 1 contacts centralia 1 and 2, distal tarsal 2 contacts centrale 2, distal tarsal 3 contacts centrale 3, distal tarsal 4 contacts centralia 3 and 4 and the fibulare, whereas distal tarsal 5 contacts the fibulare and a sesamoid bone.

Figure 7 Tarsi of different stereospondylomorph temnospondyls. (A-C) Sclerocephalus haeuseri.

(A–C) Sclerocephalus haeuseri. (A) Completely ossified tarsus (redrawn after Boy, 1988, fig. 10b, no scale bar); (B) incompletely ossified tarsus (redrawn after Krätschmer, 2004, fig. 25); (C) incompletely ossified tarsus (redrawn after Probst, 1999, fig. 25, no scale bar); (D) completely ossified tarsus of Sclerocephalus nobilis (NHMM-PW 2005/2Ls); (E) incompletely ossified tarsus of Glanochthon lellbachae (SMNS 91279); (F) incompletely ossified tarsus of Cheliderpeton vranyi (redrawn after Fritsch, 1889, plate 54). A and D are reversed to facilitate comparison with other specimens. The bones of the preaxial column are held in blue, the central column in yellow, the postaxial column in purple and the distal carpals and tarsals, respectively, in red. Abbreviations: c, centrale; dt, distal tarsal; f, fibula; fib, fibulare; int, intermedium; mt, metatarsal; se, sesamoid ossifications; t, tibia; tib, tibiale; I–V, metatarsals.

Incompletely ossified tarsi.–We are aware of two incompletely ossified tarsi of Sclerocephalus haeuseri in which the particular bones are more or less in their original position. The first one was illustrated as a photograph by Krätschmer (2004, fig. 25) and belongs to a specimen with a skull length of 143 mm (Fig. 7B). The fibula is displaced towards the tibia, so that the position of the tarsals relative to it cannot be reconstructed anymore. Five tarsal elements are ossified, the intermedium, fibulare, centralia 4 and 2 and distal tarsal 1. The particular bones are well preserved and each of them is covered by periosteal bone on its dorsal and ventral side. The largest bone is centrale 4, which is wider than long and ovate in shape. Proximal to it lies the intermedium, which is roundish in shape and approximately as long as wide. Compared to the fully ossified tarsus of S. haeuseri (see above), the fibulare is strikingly small. It is located lateral to intermedium and centrale 4 and has a square shape and is slightly longer than wide. Mediodistal to centrale 4 is centrale 2, which is wider than long and of similar size as the fibulare. Further mediodistal is a somewhat smaller, square element that is also wider than long. Although its location is closer to metatarsal II than metatarsal I, it is interpreted here as distal tarsal 1 rather than 2 because this is exactly the position of distal tarsal 1 in other specimens of S. haeuseri (see above and below) and other stereospondylomorphs as it is located in one oblique line with centralia 2 and 4 and the intermedium, and based on the data from the other specimens, it ossifies before dt2.

The second known incompletely ossified tarsus of S. haeuseri is stored at the Museum Stapf in Nierstein and was illustrated as a photograph in Probst (1999, fig. 25). The particular tarsal elements as well as tibia and fibula are slightly disarticulated (Fig. 7C). Seven tarsal bones are ossified, intermedium, centralia 4 and 2, and distal tarsalia 1, 2 and ?4. As in the other incompletely ossified tarsus, but unlike the completely ossified tarsus (see above), centrale 4 is the largest element, but it has attained a squarer shape, similar to the complete tarsus. It is wider than long. The intermedium is roundish in shape and has a somewhat concave dorsomedial margin. The fibulare is slightly larger than the intermedium. Mediodistal to centrale 4 is the broad-ovate centrale 2 whose rounded proximal margin is accommodated by the concave distal margin of centrale 4. More distally, distal tarsalia 1 and 2 are ossified, the first being roundish and the second rounded triangular in shape. A larger, reniform element is located half way between centrale 4 and metatarsal IV. It is interpreted here as distal tarsal 4, however, it could alternatively represent centrale 3 (which is nevertheless proportionally much smaller in S. haeuseri, see above).

Sclerocephalus nobilis

Completely ossified tarsus.–Schoch & Witzmann (2009, fig. 8e) illustrated and briefly described the fully ossified tarsus of Sclerocephalus nobilis. These authors did not find a tibiale and concluded that it was obscured by the tibia. Reinvestigation of the specimen (NHMM-PW 2005/2Ls) revealed a very small tibiale located mediodistal to the tibia, where it might be partially obscured by this long bone (Fig. 7D). Laterally, the tibiale forms a contact with centrale 4 and distally with the tiny centrale 1. Centrale 4 is distinctly wider than long and similar in shape and proportional size to that in adult S. haeuseri (see above). However, the intermedium differs from S. haeuseri in being proportionally larger and triangular in shape. As in S. haeuseri, the fibulare is the largest element of the tarsus, but it differs in being round-ovate in shape. The shape of the distal tarsals is either square (distal tarsal 1), triangular (distal tarsal 2), trapezoidal (distal tarsal3), irregular (distal tarsal 4) or ovate (distal tarsal 5). Interestingly, the tarsus of S. nobilis bears two small sesamoid bones exactly at the same locations as S. haeuseri.

Glanochthon lellbachae

Incompletely ossified tarsus.–From Glanochthon lellbachae, only one specimen is known that has ossified tarsal elements (Fig. 7E). This is SMNS 91281, a complete specimen with a skull length of 115 mm (Schoch, 2021), a cast of which (SMNS 91279) was studied for the present investigation. Four tarsals are ossified and aligned in an oblique straight line towards metatarsal I. The most proximal element, located between tibia and fibula, can be interpreted as the intermedium. It is comparatively small, about as long as wide and not yet differentiated in shape. Centrale 4 is somewhat larger and ovoid in shape, whereas centrale 2 is the smallest element and appears to be poorly ossified. Distal tarsal 1 is well ossified, with a large part of it being periosteal smooth bone, and has approximately the size of the intermedium.

Cheliderpeton vranyi

Incompletely ossified tarsus.–Ossified tarsal elements in Cheliderpeton vranyi have been found only in the holotype (Fritsch, 1889, pl. 54; Werneburg & Steyer, 2002, fig. 3). Whereas Fritsch (1889) designated these bones merely as Tarsalknochen (“tarsal bones”), Werneburg & Steyer (2002) interpreted the ossified elements as intermedium, tibiale, centralia 1, 2 and 4 and distal tarsal 1. In the following, we give a different interpretation of the tarsal bones, considering both Fritsch’s (1889) and Werneburg & Steyer’s (2002) illustrations (Fig. 7F). We agree with the latter authors that the intermedium is a large bone that is wider than long, rounded rectangular in shape with a convex medial margin. It articulates with the tibia medially and the fibula laterally. Mediodistal to it and distal to the tibia is a bone which was interpreted as tibiale by Werneburg & Steyer (2002). We regard this bone as centrale 4 for the following reasons. Its shape—as illustrated by Fritsch (1889)–corresponds closely to that of centrale 4 as described above in adult cf. Archegosaurus and Uranocentrodon. Furthermore, it is the largest bone of the tarsus, whereas the tibiale is a small element in all stereospondylomorphs in which it is preserved. Laterally it articulates with a bone of pentagonal shape that was interpreted as centrale 4 by Werneburg & Steyer (2002). However, its position laterodistal to the intermedium rather suggests that it is a fibulare. Our centrale 4 articulates with two further bones mediodistally and laterodistally, respectively. The small mediodistal element was interpreted as centrale 1 and the laterodistal ovate element as centrale 2 by Werneburg & Steyer (2002), but according to their positions relative to centrale 4, they probably represent centralia 2 and 3. The distalmost element is distal tarsal 1 proximal to metatarsal 1. If our interpretation is correct, then centrale 1 and the tibiale were still cartilaginous and must have been quite small elements.

Uranocentrodon senekalensis

Incompletely ossified tarsi.–From this rhinesuchid stereospondyl, only incompletely ossified tarsi are known. Van Hoepen (1915), pl. XXII) illustrated a tarsus with three ossified tarsal elements aligned in an oblique row from proximal (between the distal ends of tibia and fibula) towards metatarsal I (Fig. 8A). The most proximal bone is the largest one. It is ovate, much wider than long and is interpreted here as the intermedium. The slightly longer, but narrower centrale 4 follows mediodistally, and the small roundish centrale 2 is the distalmost element. Van Hoepen (1915, p. 144) also described more advanced tarsi with six ossified tarsal elements, but he did not illustrate them. This, however, was done by Broom (1921, fig. 2) who figured a tarsus in which intermedium, fibulare, centrale 2 and 4 as well as distal tarsals 1 and 4 are ossified (Fig. 8B). Also, in this specimen, the intermedium is the largest element, slightly larger than centrale 4. Similar to cf. Archegosaurus decheni and Sclerocephalus haeuseri, the bone is longer than wide and has a concave proximomedial margin. However, it is square rather than pentagonal in shape, and it appears to have a small notch laterally where it contacts the fibulare. Its proximal, oblique margin articulates with the mediodistal margin of the fibula. The fibulare is slightly longer than wide, with an almost straight lateral and a convex medial margin, whereas the laterodistal edge is concave (probably for articulation with the still cartilaginous distal tarsal 5, as in S. haeuseri). Centrale 4 is virtually identical in shape with that described in cf. Archegosaurus decheni (GPIT/AM/781) but is proportionally smaller. Centrale 2 is a small element that is much wider than long and ovate in shape. It articulates with the mediodistal edge of centrale 4 and with the smaller, likewise ovate distal tarsal 1. A square bone with rounded edges located proximal to metatarsal IV can best be interpreted as distal tarsal 4.

Figure 8 Tarsi of Uranocentrodon senekalensis.

(A) Incompletely ossified tarsus (redrawn after Van Hoepen, 1915, pl. XXII); (B) tarsus with more advanced ossification (redrawn after Broom, 1921, fig. 2, no scale bar). The bones of the central column are held in yellow, the postaxial column in purple and the distal carpals and tarsals, respectively, in red. Abbreviations: c, centrale; dt, distal tarsal; f, fibula; fib, fibulare; int, intermedium; mt, metatarsal; t, tibia; I–V, metatarsals.

Discussion

Degree of mesopodial ossification and body size

A noteworthy observation in the stereospondylomorph material studied here is that the degree of mesopodial ossification within the same species is not necessarily correlated with body size. Two tarsi of cf. Archegosaurus decheni, IGS U II 1/1 and IGS U II 3/1, are derived from two equally sized specimens (based on the length of the tibia and metatarsal IV) but show conspicuous differences in their extent of ossification: whereas only four tarsals are ossified in in IGS U II 3/1, there are at least eight bony elements in IGS U II 1/1. A similar observation can be made in two Sclerocephalus haeuseri specimens. Only five carpals are ossified in the hand of SMNS 90055 with a skull length of 198 mm (Schoch & Witzmann, 2009), whereas the complete carpus is ossified in SMNK-Pal 6402 with a much smaller skull length of only 150 mm. It is difficult to explain this variation in the degree of mesopodial ossification within taxa. One possible explanation might be that the specimens are derived from different populations with diverging ecologies, i.e., the specimens with more advanced ossification belonged to a population whose members were more terrestrially adapted. However, the two cf. A. decheni and S. haeuseri specimens were found within the Saar-Nahe Basin in the same stratigraphic units within the Meisenheim-Formation (M6 and M10, respectively, sensu Boy & Schindler, 2012), arguing against a different habitat and lifestyle. Another possible explanation would be that individual age rather than body size determined the degree of mesopodial ossification in these stereospondylomorphs. For example, Riquelme-Guzmán et al. (2021) reported that in the axolotl carpus the intermedium started to ossify at the age of 5 years, and in an even 20-year old specimen the tarsus was still not fully ossified. The assumption that in cf. A. decheni and S. haeuseri the largest specimens are not necessarily the oldest ones could be tested histologically by skeletchronology, but this would be beyond the scope of our paper. On the other hand, the only (partially) ossified carpus in cf. A. decheni is MB.Am.255, which according to the size of metacarpals and the zeugopodium is the by far largest specimen of this taxon (or Glanochthon latirostris).

Ossification sequence of the stereospondylomorph carpus

The small number of known ossified carpi in stereospondylomorphs allows only for a preliminary and partial reconstruction of the carpal ossification sequence (Fig. 9). The different degree of ossification in the hand of cf. Archegosaurus decheni (MB.Am.255) indicates that centralia 4 and 2 were the first elements to ossify (however, it cannot be ascertained which of the two was the first one). This is supported by the incomplete carpus of Uranocentrodon senekalensis (Van Hoepen, 1915), in which likewise only centralia 4 and 2 are ossified. In cf. Archegosaurus, this is followed by ossification of the intermedium, radiale, centrale 1 and distal carpal 1 (MB.Am.255). The only known incompletely ossified carpus of Sclerocephalus haeuseri (Schoch & Witzmann, 2009) contains all these bones with the exception of the intermedium (Fig. 9); thus the latter bone may have started to ossify after radiale, centrale 1 and distal carpal 1 also in cf. Archegosaurus. The only known carpus that is ontogenetically more advanced is that of S. haeuseri specimen SMNK-Pal 6402 in which all elements with the exception of centrale 3 are present (Fig. 9). However, this does not necessarily mean that centrale 3 was the last element to ossify; if it was a similarly small element as centrale 3 in the tarsus, then it may simply be not preserved.

Figure 9 Preliminary ossification sequence of carpal elements based on cf. Archegosaurus decheni, Sclerocephalus haeuseri and Uranocentrodon senekalensis.

Ossified elements of the preaxial column are represented by blue boxes, of the central column in yellow boxes, of the postaxial column in purple boxes and of distal carpals in red boxes. White boxes represent cartilaginous elements. Abbreviations: c, centrale; dc, distal carpal; int, intermedium; rad, radiale; uln, ulnare.

Considering this admittedly very small set of data, a tentative and very preliminary ossification sequence of carpal elements based on cf. Archegosaurus, S. haeuseri and Uranocentrodon is as follows (the sequence within the square brackets is unclear): (centrale 4, 2) → (radiale, centrale 1, distal carpal 1) → (intermedium) → (distal carpals 2–4, ulnare). In conclusion, considering the scheme of Goette (1879), the central column of the mesopodium starts to ossify first in the stereospondylomorph carpus, followed by the preaxial column. The postaxial column starts to ossify last. Further finds of temnospondyl and especially stereospondylomorph carpi are necessary to confirm or reject this sequence and to hopefully add more data. Additionally, this sequence is weakened by the fact that it is based on three different stereospondylomorph taxa and not on a single species.

Ossification sequence of the stereospondylomorph tarsus

The ossification sequence of tarsal elements can be reconstructed with more confidence and at a higher resolution than the carpus because a larger number of specimens with ossified tarsal elements is preserved (Fig 10). However, it is still not possible to provide a complete sequence. Most data are derived from cf. Archegosaurus decheni. Based on the drawings of von Meyer (1858) and Witzmann (2006) as well as specimens IGS UII 1/1 and GPIT/AM/781, the following sequence for cf. A. decheni can be reconstructed (the sequence within the square brackets cannot be resolved) (Fig. 10): (intermedium) → (centralia 4, 2) → (distal tarsal 1) → (fibulare, distal tarsalia 2–4) → (tibiale, centralia 1, 3, distal tarsal 5). In Sclerocephalus haeuseri, less tarsi are preserved, but may provide supplementary data to cf. Archegosaurus (Fig 10). The specimen illustrated by Krätschmer (2004) shows that the fibulare ossified prior to distal tarsalia 3-4. The more advanced specimen illustrated by Probst (1999) may indicate that distal tarsals 2 and 4 ossified prior to distal tarsals 3 and 5. This is supported by Broom (1921) who described two Uranocentrodon specimens in which among the distal tarsals only the first and fourth were ossified (Fig 10). In this respect it might be interesting to note that among distal tarsals, the fourth one is often the largest in early tetrapods (Carroll & Holmes, 2007), and this is pronounced in S. haeuseri (see above). Maybe this can be correlated with the earlier ossification of that bone, but hopefully further fossil finds will clarify if this sequence is real or an artefact. Further refinement of the stereospondylomorph tarsal ossification sequence is allowed by Cheliderpeton vranyi as illustrated by Fritsch (1889) which shows that the fibulare ossified prior to distal tarsals 2–4 (Fig 10). However, it has to be kept in mind that there is some degree of uncertainty with respect to the identity of tarsal elements in this specimen. Combining these data, a preliminary stereospondylomorph tarsal ossification sequence can be reconstructed as follows: (intermedium) → (centralia 4, 2) → (distal tarsal 1) → (fibulare). Then the sequence appears to differ between taxa. Based on cf. Archegosaurus, Uranocentrodon and S. haeuseri, the sequence continues as follows: (distal tarsal 4) → (distal tarsal 2 (not resolvable in cf. Archegosaurus)) → (tibiale, centralia 1, 3, distal tarsals 3, 5). In Cheliderpeton vranyi, however, centrale 3 ossified prior to distal tarsals 2, 4 and 5, and before the tibiale and centrale 1. In conclusion, the central column of the mesopodium starts to ossify first in the stereospondylomorph tarsus, followed by the postaxial column, whereas the preaxial column is always the last to ossify. As with the ossification sequence of the carpus, we have to admit that, this sequence is weakened by the fact that it is based on different stereospondylomorph taxa and not on a single species.

Figure 10 Preliminary ossification sequence of tarsal elements based on cf. Archegosaurus decheni, Sclerocephalus haeuseri, Uranocentrodon senekalensis and Glanochthon lellbachae.

Ossified elements of the preaxial column are represented by blue boxes, of the central column in yellow boxes, of the postaxial column in purple boxes and of distal carpals in red boxes. White boxes represent cartilaginous elements. Abbreviations: c, centrale; dt, distal tarsal; fib, fibulare; int, intermedium; tib, tibiale.

Comparison between the patterns of carpal and tarsal ossification in stereospondylomorphs

Some interesting differences between the ossification sequence of the stereospondylomorph carpus and tarsus exist. In the carpus, the preaxial column (radiale and centrale 1) starts to ossify earlier than the postaxial one, whereas in the tarsus, it is vice versa with the preaxial side (tibiale and centrale 1) being the last column to ossify. Furthermore, the intermedium is the first element to ossify in the tarsus at least in cf. Archegosaurus, which implies a subsequent proximal to distal polarity during ossification in the central and postaxial columns. In contrast, the intermedium ossifies much later (as the sixth element) in the carpus, so that a partial distal to proximal polarity is present.

Despite these differences, a general ossification pattern that starts from proximolateral (intermedium or centrale 4) to mediodistal (distal tarsal and carpal 1) roughly in a diagonal line is common to all mesopodials investigated. This was already observed by Schaeffer (1941, p. 400) based on the illustration of a cf. Archegosaurus foot in von Meyer (1858, pl. XIX, fig. 8) mentioned above (Table 1). He suggested that this pattern might basically reflect the alignment of stress within the mesopodium during locomotion.

Comparison with other temnospondyls, lissamphibians and amniotes

Jia et al. (2022) recently conducted a large-scale analysis of the mesopodial ossification patterns in a great number of salamander taxa in a variety of clades. They found preaxial polarity in the ossification sequence of the distal carpals and tarsals (i.e., the digital arch mesopodials) in most clades of salamanders, whereas others, like plethodontids and salamandrids, show postaxial polarity. In contrast, the non-digital mesopodial elements show postaxial polarity in all salamander taxa as known in anurans and amniotes, i.e., the preaxial column with radiale/tibiale and element y is the last to ossify in salamanders. In some taxa, ossification in both the preaxial and central column proceeds in a proximo-distal direction, in others it is vice versa. This variation in the ossification of mesopodial elements in salamanders is intriguing and a certain degree of variation seems to be reflected also in the data at hand on mesopodial ossification in Paleozoic stereospondyls. Both datasets, the extensive one on salamanders compiled by Jia et al. (2022) and the data on stereospondyls in this study, pertain to the latest stages of limb development, i.e., the ossification of mesopodial elements.

Interestingly, while the general features of earlier events in skeletal development of salamanders, i.e., the very early condensation of the basale commune (distal carpal/tarsal 1 + 2) and digits I and II, as well as an overall preaxial polarity in zeugopod and digit development, are conserved in salamanders, there is notable variation in the earlier stages of skeletal development in salamander autopods during mesenchymal condensation, specifically in the region of the forming mesopod (Shubin & Wake, 2003) (Fig. 10). Four main areas of mesenchymal fields within the developing salamander autopod have been identified: the preaxial column, the central series, the postaxial column, and the digital arch (Blanco & Alberch, 1992; Shubin & Alberch, 1986; Wake & Shubin, 1998) and variation is generated though the size of mesenchymal fields and different patterns of connection between these four regions.

The early skeletal development of the limbs of seven salamander taxa have been studied to date in enough detail in this respect to allow for comparisons between them, namely Salamandrella keyserlingii (Schmalhausen, 1910; Vorobyeva et al., 2000; Vorobyeva & Hinchliffe, 1996), Ranodon sibiricus (Schmalhausen, 1917; Vorobyeva, Ol’shevskaya & Hinchliffe, 1997), Salamandra salamandra (Strasser, 1879), Dicamptodon ensatus (Wake & Shubin, 1998), Triturus marmoratus (Blanco & Alberch, 1992), Ambystoma mexicanum (Shubin & Alberch; 1986, Steiner, 1921), Desmognathus aenus (Franssen et al., 2005) and Plethodon cinereus (Kerney, Hanken & Blackburn, 2018) (Fig. 11). Two further taxa, Proteus anguinus (Shubin & Alberch, 1986) and Amphiuma means (van Pée, 1904), have strongly reduced limbs, which somewhat hampers a detailed comparison with taxa that retain the full complement of digits. Nevertheless, the available data suggest some general patterns in the formation of the mesenchymal field. The preaxial column is particularly pronounced in Ambystoma and Triturus both of which have pond-dwelling larvae with a comparatively long larval development. Contrary, Dicamptodon ensatus, a highly terrestrial species as an adult, and plethodontid salamanders have a pronounced central column. The hynobiids Salamandrella and Ranodon show an almost simultaneous development of all three columns (Schmalhausen, 1910; Vorobyeva & Hinchliffe, 1996; Vorobyeva, Ol’shevskaya & Hinchliffe, 1997; Vorobyeva & Mednikov, 2004). Considering the rather basal position of hynobiids in the salamander phylogeny (Wiens, Bonett & Chippindale, 2005; Zhang & Wake, 2009; Jia et al., 2022), this may indicate a more plesiomorphic pattern of development, but this needs to be investigated in more detail in the future. Salamanders are a large vertebrate clade with close to 800 species in 10 families, which display an impressive diversity of habitats, lifestyles, and life history patterns (www.amphibiaweb.org). Considering this great breadth of ecologies and life history patterns, it may not be too surprising to see variation in the specific patterns of limb development. Life history has previously been suggested to play a role in the pronounced differences in overall timing between fore- and hind limb formation in different salamander taxa (Shubin & Wake, 2003). Therein salamanders with free swimming larval stages, such as the axolotl, hatch from the egg with the forelimb in limb bud stage and the hind limb still entirely absent (Nye et al., 2003). The delay between the onset of fore- and hindlimb formation may be as long as several weeks in these taxa (Blanco & Alberch, 1992; Nye et al., 2003). Contrary, fore- and hindlimb development take place almost simultaneously in direct developing taxa such as Desmognathus aenus, which hatch from the egg as miniatures of the adults with fully formed limbs.

Figure 11 Patterns of early mesenchymal formation of limb skeleton in selected salamander taxa in the four units of the developing limb (see Shubin & Wake, 2003).

Development proceeding from left to right. Small black arrows in schematic drawing on the far right indicate direction in which mesenchymal fields are expanding and forming. Data based on Franssen et al. (2005) for Desmognathus aenus, van Pée, (1904) for Amphiuma means, Shubin & Alberch (1986) for Proteus anguinus, Blanco & Alberch (1992) for Triturus marmoratus, Strasser (1879) for Salamandra salamandra, Shubin & Alberch (1986), Steiner (1921) and Nye et al. (2003) for Ambystoma mexicanum, Wake & Shubin (1998) for Dicamptodon ensatus, Schmalhausen (1917) and Vorobyeva, Ol’shevskaya & Hinchliffe (1997) for Ranodon sibiricus, Schmalhausen (1910), Vorobyeva et al. (2000) and Vorobyeva & Hinchliffe (1996) for Salamandrella keyserlingii, personal observations for Cryptobranchus alleganiensis. Phylogenetic backbone following the tree topology by Zhang & Wake (2009).

The sequence of ossification in the stylopod, zeugopod and the digits broadly reflects the overall patterns of pre- vs. postaxial development in anamniote tetrapods, but without an exact recapitulation of the earlier events in skeletogenesis (Fröbisch, 2008). Therein, the delayed ossification of the mesopodial elements represents the most obvious deviance from earlier patterns. Likewise, there is no obvious direct correlation between the patterns of ossification in the salamander mesopod as described by Jia et al. (2022) and the very early patterns of mesenchymal fields in the mesopodium outlined above (Fig. 11). Currently, the potential linkages between very early events in limb skeletal formation of salamanders and later patterns of ossification and as well as any connections thereof to specific ecologies, life history patterns or other biological factors remain insufficiently understood to draw more detailed conclusions and there is no obvious linkage that readily stands out between the early patterns of skeletogenesis and life history and/or phylogeny (Fig. 11). Nevertheless, the variation in the time and dominance of the mesenchymal columns in the forming mesopodial regions of salamanders and the later pattern of variation in ossification is intriguing and it is likely that skeletal development in Paleozoic and Mesozoic anamniotes followed an overall similar principle (Fröbisch, Carroll & Schoch, 2007; Fröbisch, 2008; Fröbisch, Bickelmann & Witzmann, 2014; Fröbisch et al., 2015). Jia et al. (2022) compared their results with published descriptions of incompletely ossified carpals and tarsals of stem-tetrapods, stem-amniotes and selected temnospondyls. They found preaxial polarity in the digital mesopodial arch of the stem-tetrapods Acanthostega and Greererpeton, in the presumed stem-amniote Proterogyrinus, and in the amphibamiform temnospondyls Amphibamus and Gerobatrachus, whereas in the stem-tetrapod Whatcheria the development is probably postaxial, as it is in anurans and amniotes. Interestingly, Jia et al. (2022) found postaxial polarity in the ossification of the non-digital arch mesopodium in all above mentioned early tetrapods plus the stem-tetrapods Panderichthys and Ossinodus. These authors thus considered preaxial dominance in the ossification of distal carpals and tarsals as plesiomorphic for tetrapods, thus confirming the hypothesis of Fröbisch, Carroll & Schoch (2007) and Fröbisch et al. (2015), whereas postaxial dominance is plesiomorphic for the remaining mesopodial elements.

Our study of the ontogeny of the mesopodium in stereospondylomorph temnospondyls confirms Jia et al.’s (2022) conclusion about tarsal development: the non-digit mesopodials show postaxial dominance, i.e., the preaxial arch consistently starts to ossify after the central and postaxial columns (however, it could not be resolved if the tibiale or centrale 1 ossified first within the preaxial column). Also, in accordance with most early tetrapods and salamanders, the distal tarsals in the investigated stereospondylomorphs show preaxial development; however, in Uranocentrodon and Sclerocephalus haeuseri, distal tarsal 4 probably ossified after distal tarsal 1, and followed by distal tarsals 2, 3, and 5. If this observation is correct and not based on artefacts of preservation, this would represent a variation in preaxial polarity of distal mesopodials not known in extant forms.

The development of the stereospondylomorph carpus, however, differs from the above described pattern. Whereas the development of distal carpals is preaxial as in most early tetrapods (for which a statement can be made), the non-digit mesopodials show preaxial development, i.e., the preaxial column ossified after the central column and before the postaxial one. This pattern is unique among the known early tetrapods and occurs only in certain salamanders (Salamandridae) (Jia et al., 2022). Furthermore, ossification proceeds from distal to proximal in the central column of the stereospondylomorph carpus.

Conclusions

Although our results concerning the ossification patterns of the stereospondylomorph carpus and tarsus are based on a relatively small number of specimens and the investigation is often hampered by poor preservation of the mesopodia, especially in the carpus, our observations might point to a greater variability in the development of the mesopodium in stereospondylomorphs and probably other early tetrapods, possibly mirroring a similar variation as seen in the early phases of skeletogenesis in salamander carpus and tarsus. To get a broader picture of the development of the mesopodium in early tetrapods it would be desirable to learn more about its ossification pattern in non-stereospondylomorph early tetrapods. However, data are limited by the lack of adequate growth series in which the ossification sequence of carpals and tarsals can be traced. This will hopefully be overcome by new fossil finds in the future and complemented by a more detailed understanding of the effects of life history and ecology on limb skeletal development in extant tetrapods.

David Dilkes and two anonymous referees are thanked for their thoughtful reviews. We are grateful to Ingmar Werneburg (Tübingen) for the loan of GPIT/AM/781a, b. Markus Brinkman (Berlin) is thanked for producing casts of GPIT/AM/781a, b and MB.Am.255a, b. Jean-Claude Horrenberger (Strasbourg), Herbert Lutz (Mainz), Wolfgang Munk (Karlsruhe) and Rainer Schoch (Stuttgart) are thanked for access to the collections under their care. We thank the following colleagues for providing photographs of fossil mesopodia: Manuela Aiglsdorfer (Mainz) and Rainer Schoch (Stuttgart), Christiane Birnbaum, Tim Niggemeyer, and Mathias Vielsäcker (Karlsruhe), Celeste Pérez-Ben (Berlin) and Kévin Janneau (Strasbourg).

Additional Information and Declarations

Competing Interests

Author Contributions

Data Availability

The authors declare that they have no competing interests.

Florian Witzmann conceived and designed the experiments, performed the experiments, analyzed the data, prepared figures and/or tables, authored or reviewed drafts of the article, and approved the final draft.

Nadia Fröbisch conceived and designed the experiments, performed the experiments, analyzed the data, prepared figures and/or tables, authored or reviewed drafts of the article, and approved the final draft.

The following information was supplied regarding data availability:

The data is in the Material and Methods section and in the tables. All the other data is included in the figures.

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
