# Peer review of "Morphology and ontogeny of carpus and tarsus in stereospondylomorph temnospondyls"

_PeerJ, doi:10.7717/peerj.16182_

## Round 0.1 · original submission · Minor Revisions

Apologies for the delay in obtaining reviews. However, I am happy to say that all three reviewers really like the paper and give constructive critiques. There are just some relatively minor issues needing amendment; largely in wording, We look forward to the revised paper.

·

Basic reporting

No comment.

Experimental design

No comment.

Validity of the findings

No comment.

Additional comments

General Comments.
Thank you for the opportunity to read this manuscript. The carpus and tarsus of early tetrapods is an often-neglected area of their anatomy. This paper adds significantly to our knowledge of this anatomical region in stereospondylomorphs and provides fascinating insights into patterns of development with implications for the evolution of early and modern amphibians. Overall, an important contribution to the study of amphibian evolution.

Specific Comments.
Lines 396-397: The authors state centrale 4 of cf. Archegosaurus decheni (Fig. 5) differs from the rectangular-shaped centrale 4 of Eryops in having a longer medial than lateral side. However, this difference in the relative lengths of the two sides of centrale 4 is also true for Eryops, Acheloma, and Dissorophus where the side facing the tibiale is longer than the side facing the fibulare. The difference is not as marked as in this individual of cf. A. decheni, hence the description of centrale 4 as rectangular in Eryops, Acheloma, and Dissorophus, but it is a shared feature of these genera.

Lines 397-398: The proximal and distal sides of centrale 4 in Acheloma are concave as in this individual of cf. A. decheni which accounts for the smaller length of the side facing the fibulare. The proximal side of centrale 4 in Eryops is concave, but the distal side has only a modest (and difficult to see) concavity where it contacts distal tarsal 4. It’s uncertain if the same is true for Dissorophus, but I suspect it is because of the similar difference in the relative lengths of the medial and lateral sides.

Line 398: It’s interesting that the dorsal side of centrale 4 in this individual of cf. A. decheni has a concave dorsal side whereas it is the ventral side that is deeply concave in Acheloma, Eryops, and Dissorophus.

Line 404: Another possible indication of an incompletely ossified tibiale is the presence of an apparent contact surface along the entire lateral side of centralia 4 that, using Eryops and Acheloma as guides, contacted the tibiale. Since the length of this contact surface is greater than the preserved length of the tibiale, the tibiale was likely longer than preserved.

Line 406: This individual of cf. A. decheni appears to have a more substantial contact between centrale 2 and distal tarsal 3 than in Eryops and Acheloma.

Lines 408-409: I believe centrale 1 contacts the tibiale proximally rather than distally.

Line 428: The word were in “If we assume a boundary in the region were …” should be replaced with where.

Line 534. S. haeuseri should be italicized.

Reviewer 2 ·

Basic reporting

This is a very polished manuscript with just a few typos to correct. The language is clear and unambiguous and the descriptions are well structured and easy to follow. The adequate literature is cited, hypotheses are well explained and the results and discussion are sufficiently illustrated with high quality figures.

Experimental design

The experimental design is adequate and I find no fault in the general hyptheses put forward and the way these were addressed. Autopodial development is an area with a long standing interest among morphologists and the obvious differences between salamanders and other tetrapods with regards to autopodial have long been puzzling. Data from the fossil record have recently shown that the pattern is not as clear cut as previously thought and the manuscript by Witzmann and Fröbisch provides important data that will help frame this discussion.
There are some patterns with regard to degree of ossification and specimen size that are somewhat puzzling, as flagged by the authors. Doing skeletochronology might indeed help with these, but there are plenty of other hypotheses to explain the size-degree of ossification mismatch and given the overall rarity of the material I would consider it unwise to insist on a destructive sampling of the specimens for a skeletochronological analysis.

Validity of the findings

no comment

Additional comments

line 66: delete third "and"
line 122: substitute "which" with "that"
line 543: italicise "S. haeuseri"
line 816: correct broader

Reviewer 3 ·

Basic reporting

This manuscript is well written and has a clear aim which is met through the analysis of the data at hand.
It is mostly a synopsis of existing material, which has been reanalyzed and redrawn for the current study. I believe it has covered the relevant literature for the field, as I have not seen any egregious oversight. The references in table 1 are thoroughly researched.

The first two paragraphs of "Materials and Methods" could be turned into a single table.

It would also be helpful to see a more authoritative reference than Carroll 1997 on the formation of periosteal mesopodial bone before endochondral. This is the opposite of most long bones, and the claim should reference specific studies. Marissa Fabrezi has published a lot on anuran mesopodial development, and may have a solid primary reference. There may also be a primary reference in Carroll's book.

There is only one minor reporting error on line 816 where the word "braoader" has an extra a.

Experimental design

I believe that there should be a paragraph in the introduction that guides the non-paleontologist reader through the assumptions of this analysis. The presence/ absence of tiny bones in ancient mesopodia requires much interpretive license to make conclusions about ontogeny. Taphonomic loss, cryptic species, and even spurious false positives are all potential sources of error. The authors are left to make parsimonious conclusions about the material at hand. This may be the standard in looking at ontogenetic series in fossils and could be received wisdom in a paleontological publication. However, deciphering these assumptions is distracting to this neontologist reader who has struggled enough with discerning developmental patterns from extant material.

Validity of the findings

The authors place the necessary caveats about the patterns of temnospondyl ossification in their conclusion. These need to be drawn from disparate taxa and reveal counterintuitive trends.

I applaud the author's review of salamander condensation studies in figure 11 and their caution in drawing any rash conclusions about life history or habitat beyond those speculations from the preexisting literature. Have any of these references examined the patterns of ossification as they relate to condensation patterns? I don't think any have but I may be wrong. It would be a helpful analysis to support/ refute the view that mesopodial ossification patterns do not necessarily follow limb bud condensation patterns.

---

## Round 0.2 · accepted · Accept

The revisions have been thoughtfully handled. I am happy to recommend publication -- congratulations!!

p14 "contact surface for the tibiale along the complete lateral side of centrale 4 that is longer than the preserved length of the tibia" -- be sure to correct this in proof (or sooner)!! "tibiale", not "tibia"!!!!

Thanks for sending this excellent paper to PeerJ. It's a fabulous contribution to the journal.